# Diagnosis and Management of UTI in Primary Care Settings—A Qualitative Study to Inform a Diagnostic Quick Reference Tool for Women Under 65 Years

**DOI:** 10.3390/antibiotics9090581

**Published:** 2020-09-07

**Authors:** Emily Cooper, Leah Jones, Annie Joseph, Rosie Allison, Natalie Gold, James Larcombe, Philippa Moore, Cliodna A. M. McNulty

**Affiliations:** 1Primary Care and Interventions Unit, Public Health England, Twyver House, Bruton Way, Gloucester GL1 1DQ, UK; rosie.allison@phe.gov.uk (R.A.); Cliodna.McNulty@phe.gov.uk (C.A.M.M.); 2Nottingham University Hospitals, Hucknall Rd, Nottingham NG5 1PB, UK; Amelia.Joseph@nuh.nhs.uk; 3Public Health England Behaviour Insights (PHEBI) Team and University of Oxford, Wellington Square, Oxford OX1 2JD, UK; natalie.gold@phe.gov.uk; 4General Practitioner, County Durham, UK; terriandjames@btinternet.com; 5Gloucestershire Hospitals National Health Service Foundation Trust, Great Western Rd, Gloucester GL1 3NN, UK; philippa.moore1@nhs.net

**Keywords:** antibiotic prescribing, antimicrobial resistance, survey, questionnaire, knowledge, expectations, antibiotics, delayed prescribing, *Escherichia coli* bloodstream infection, EBSI, bloodstream infection, BSI, pyelonephritis, behavioural science, general practice, diagnostic

## Abstract

**Background:** To inform interventions to improve antimicrobial use in urinary tract infections (UTIs) and contribute to a reduction in *Escherichia coli* bloodstream infection, we explored factors influencing the diagnosis and management of UTIs in primary care. **Design:** Semi-structured focus groups informed by the Theoretical Domains Framework. **Setting:** General practice (GP) surgeries in two English clinical commissioning groups (CCGs), June 2017 to March 2018. **Participants:** A total of 57 GP staff within 8 focus groups. **Results:** Staff were very aware of common UTI symptoms and nitrofurantoin as first-line treatment, but some were less aware about when to send a urine culture, second-line and non-antibiotic management, and did not probe for signs and symptoms to specifically exclude vaginal causes or pyelonephritis before prescribing. Many consultations were undertaken over the phone, many by nurse practitioners, and followed established protocols that often included urine dipsticks and receptionists. Patient expectations increased use of urine dipsticks, and immediate and 5 days courses of antibiotics. Management decisions were also influenced by patient co-morbidities. No participants had undertaken recent UTI audits. Patient discussions around antibiotic resistance and back-up antibiotics were uncommon compared to consultations for respiratory infections. **Conclusions:** Knowledge and skill gaps could be addressed with education and clear, accessible, UTI diagnostic and management guidance and protocols that are also appropriate for phone consultations. Public antibiotic campaigns and patient-facing information should cover UTIs, non-pharmaceutical recommendations for “self-care”, prevention and rationale for 3 days antibiotic courses. Practices should be encouraged to audit UTI management.


**Strengths and limitations of this study:**
(1)By purposively selecting CCGs with different sociodemographic characteristics and then inviting surgeries to participate in a random order, we were able to reduce bias and get input from surgeries with a range of antibiotic prescribing rates.(2)We were able to speak with 57 staff members from 8 practices with various roles in the management pathway. This allowed us to explore the behaviour of non-prescribers.(3)General practices that were more overburdened or going through changes were more likely to decline to participate, and their views may not have been captured.(4)Using the Theoretical Domains Framework to develop our focus group schedule allowed us to explore all areas of behaviour and fed into intervention development.(5)Nesting this piece of research within a project that included the development of UTI resources for all ages, including a UTI flowchart and leaflet for older adults, could have biased discussion to focus more on management in older adults (which is seen as more complex by clinicians) even when general feedback was prompted for during the discussion.


## 1. Introduction

*Escherichia coli* is the main cause of bloodstream infection in the UK and is responsible for more than one-third of bloodstream infection cases in England each year [1,2]. Yearly rates of *E. coli* bloodstream infections (*EC*BSIs) increased by 12% between 2014 and 2018 (65.7/100,000 population in 2014 to 76.0/100,000 population in 2018) [3]. Modelling, using English mandatory surveillance data in 2017, showed that if rising trends are not influenced, there will be a 5.1% year-on-year increase in incidence of *EC*BSIs by 2020/21 [4]. This will be even higher for community onset cases (7.8%) [4].

Because most cases of *EC*BSI (51% of cases) are attributed to an underlying urological condition, infections of the urogenital tract contribute the highest number of deaths when comparing 30 days all-cause mortality in people with *EC*BSI [2,5]. Independent risk factors for *EC*BSI related to the urogenital tract include experiencing or having treatment for a UTI in the previous month, having a short- or long-term urinary catheter, and being a woman [2,6,7,8].

We may be able to prevent and reduce *EC*BSIs if we effectively diagnose and manage community acquired UTIs in adults [9]. Antimicrobial stewardship policies and interventions over the last decade have focused on improving management of common infections presenting in primary care [10,11]. However, key challenges of inappropriate prescribing and diagnosis remain for UTI and overuse has continued to drive antimicrobial resistance in Gram-negative organisms causing UTIs and BSIs. One-fifth of antibiotic prescriptions between 2013 and 2015 in England were for UTI-related conditions [12]. In 2014, English UTI antibiotic guidelines changed to suggest nitrofurantoin as first-line treatment for lower UTI, and in 2016/17, a reduction in the trimethoprim:nitrofurantoin prescribing ratio was included in the Quality Premium [13]. This has probably contributed to the significant decreases in trimethoprim use (by 41% since 2014) and a decrease in *E. coli*-resistant trimethoprim within UTI samples in England (35.1% to 31.2% between 2015 and 2018) [3]. However, we know that these improvements have varied significantly between health authorities across England and more needs to be done to support the universal use of national prescribing guidance [14]. Greater trimethoprim prescribing (compared to nitrofurantoin) is associated with a higher incidence of BSI, UTI-related trimethoprim-resistant *E. coli* BSI, and increased incidence of trimethoprim-resistant bloodstream infection [9].

To help inform interventions to improve the management and diagnosis of UTIs within primary care, the authors explored GP staff’s diagnosis and management of UTI in adults under 65 years. This coincided with a planned review of the 2017 Public Health England (PHE) diagnosis of urinary tract infections: quick reference tool for primary care [15]. Specific aims of this review included:Determine the practices of primary health care staff when diagnosing and treating UTIs in patients under 65 years in the primary health care setting,identify where there are current barriers and facilitators to appropriate diagnosis and treatment, andidentify how these gaps may be addressed through resource development or other measures.

Although the aims of the review included the whole population of England, large parts of the focus group discussions were focused on women under 65 years, and this is the focal group for the qualitative findings highlighted in this paper.

## 2. Methods

We aimed to attain views from primary health care staff with a range of experience around the diagnosis and management of UTI in England. Therefore, we purposively sampled in two areas with different demographics/populations: Gloucestershire and Nottingham City (CCGs). These two clinical commissioning groups (CCGs) were chosen as they provided different types of urban/rural, socioeconomic and ethnic population breakdowns based on information from the 2011 UK national census (Table 1) [16].

CCGs commission medical care for GP practices, deciding what services are needed for diverse local populations, and ensuring that they are provided. Most people who reside in the UK have access to primary care through a GP practice. Practices vary in their rates of urine submission to local laboratories, and this may be associated with differing staff views on the management of UTI. To account for this, practices in each CCG were stratified by higher, middle or lower rates of local laboratory urine specimen submission. To try and reduce sampling bias and to avoid only enrolling practices with an interest in UTIs, GP practices in each stratum were randomised and then approached by letter and telephone in random order until one practice per strata in each region was recruited.

Invitation letters from PHE contained study information sheets and a consent form. Each participant was offered a £20 high-street voucher incentive to participate. Response rates were 25% (*n* = 16/4) and 24% (*n* = 17/4) in the Gloucestershire and Nottingham City CCGs, respectively. Data was collected from June 2017 to March 2018 (Figure 1).

Focus group discussion schedules were developed by the research group, including 2 microbiologists, a patient, a psychologist, researchers and a GP, and were informed by the Theoretical Domains Framework (TDF) (Table 2) [21]. Topic areas included primary care staff attitudes and experiences of managing suspected and proven urinary tract infections, antimicrobial resistance and antibiotics, and a discussion with staff about their thoughts and opinions on the diagnostic process proposed in the draft UTI diagnostic flowchart. The focus group discussion schedule for the practices was piloted, but there were no significant changes made following the pilot and findings from this group are included in the results.

Focus groups were facilitated and observed by trained qualitative researchers, held in general practices, and lasted from 30 to 90 min. Discussions were audio-recorded, transcribed verbatim and checked for accuracy by a member of the research team.

Data was analysed inductively by PHE researchers, LJ and RA, using QSR Nvivo 10 [22,23]. Themes were refined until redundant or infrequent codes were recoded. Ten percent of transcripts were double coded by a second researcher (NG). Codes were discussed by the researchers, and a consensus was reached. After this initial analysis, the results with representative participants’ quotes were discussed by the research group and at an expert stakeholder workshop attended by GPs, nurses, pharmacists, public representatives, microbiologists, and other medical professionals. The themes were then considered within the TDF framework. After each focus group, GP staff’s opinions were used to tentatively modify a UTI quick reference diagnostic tool for the next focus group. Quotes specific to patients over the age of 65 years were coded apart from those from the discussion about general patients and are presented elsewhere [24]. Though the general discussion did at times include discussion around men and a children, most of the focus was on younger women presenting at the practice as this was the focus of the UTI diagnostic tool.

Quotations from the transcripts are used in the results table to illustrate the most prominent TDF domains (Table 3) and findings. The terms used to explain how many groups agreed on or discussed an issue or finding in the results include “most”—meaning that 4 or more focus groups raised or discussed the issue; or “some” meaning that 1 to 3 groups raised or discussed the issue.

Data was triangulated through discussions during an expert stakeholder workshop discussed previously (January 2018), where experts discussed and applied the findings to recommendations for diagnostic tool development.

## 3. Consultations

### 3.1. Role and Professional Identity

Findings indicated that management of UTI in younger patients was shared by most staff within the surgery. Receptionists in some surgeries asked patients who telephone with urinary symptoms to bring urine samples into the surgery and provided guidance on sample collection. Nurses often tested urine samples left by patients at reception and referred the patients to clinicians if they were concerned the patient had a UTI. Nurse prescribers frequently managed patients with uncomplicated cases of UTI, without any referral to other medical staff. GPs reported having face to face consultations for UTIs and telephone consultations, managing complicated cases and treatment failure.

### 3.2. Role and Professional Identity/Environmental Context and Resources

Almost all focus groups reported that they may use telephone consultations to manage patients who have a suspected UTI. At the telephone consultation, some clinicians asked for a urine sample to be brought in but others considered this as unnecessary. Patients dropping unrequested urine samples off at GP reception was reported by most groups as a challenge in diagnosing and managing UTIs, and some groups reported that they would then have to ring the patient for a clinical history. Some groups reported that they would prescribe antibiotics over the phone for a patient with 2–3 urinary symptoms, only requesting clinical examination to rule out pyelonephritis if: the patient had recurrent UTIs, there were any risk factors for more severe infection, or the clinician was worried about the patient.

### 3.3. Diagnosis

#### 3.3.1. Skills

Three groups reported asking about vaginal discharge and considering other genitourinary causes during a consultation for suspected UTI. Some of these groups indicated that other genitourinary causes were considered later in the consultation after ruling out a UTI (by using urine dipstick).

There was a range of opinions on when to send a urine for culture and when to rely on urine dipsticks for patients. Most groups said they would send a culture following treatment failure or for a recurrent UTI. Some groups reported sending a urine for culture if the dipstick was negative but the patient was symptomatic, if the patient was quite unwell, for male patients, for pregnant women, and for children. Some groups indicated that they would not send a urine culture if the dipstick was negative or if the symptoms clearly indicated an uncomplicated UTI.

#### 3.3.2. Role and Professional Identity/Social Influence

Most groups indicated that urine would always be dipped if it was dropped at reception or brought in with the patient who had a suspected UTI. Sometimes, this was at the request of the nurse/receptionist and performed by the nurse. Some groups felt that patients expected that a dipstick would be used as part of the diagnostic process. This expectation and readily available urine samples influenced their decision to use urine dipsticks as part of the diagnostic process, though some prescribers indicated that they would not ask for a sample if the patient was strongly symptomatic (with three symptoms, as per the 2017 PHE UTI diagnostic guidance) [15]. Participants also reported varying interpretation and use of urine dipsticks, especially when symptoms did not match the urine dipstick results.

#### 3.3.3. Environment Context and Resources

Most groups reported using the 2017 PHE UTI diagnostic guidance or local guidance to guide decision making, though they might not refer to it every time [15] Participants reported that laboratory specimen collection services were not frequent enough to allow culture results to inform treatment. Some groups suggested that an accurate point-of-care test to aid in the diagnosis of a UTI would be very useful. This was often related to the clinician’s desire to have a visible test with which to reassure the patient, or a result to reliably determine the need for antibiotics.

#### 3.3.4. Beliefs about Consequences/Environmental Context and Resources

Some groups indicated that concern that they might miss sepsis/pyelonephritis caused them to be overly cautious when diagnosing and treating UTIs. When reviewing the ideas for the updated diagnostic tool, most agreed that the exclusion of pyelonephritis and sepsis needed to be included, as part of any updates. There was discussion about what stage pyelonephritis and sepsis should be considered in a UTI consultation; some groups thought it should be excluded early on and others thought it should be ruled out after UTI symptoms were discussed.

### 3.4. Medical Management

#### 3.4.1. Environmental Context and Resources

Some groups expressed the need for more or clearer national guidance advising about key areas of the management of UTIs. This included recurrent UTIs (three groups), catheterised patients (one group), and frank haematuria (one group).

#### 3.4.2. Reinforcement

Most groups expressed the belief that patients expect to be prescribed an antibiotic if they have urinary symptoms, especially if they have had them before or if the dipstick is positive. This expectation was reinforced by being given antibiotics at previous consultations for similar symptoms.

#### 3.4.3. Memory, Attention, and Decision Making/Social Influence/Optimism

Most groups would not usually use a delayed/back-up prescription for UTIs. Some considered this option when waiting for results from urine culture if symptoms were mild. Practices generally reported that patients did not understand that a UTI can safely clear on its own and they expected antibiotics. Some groups felt that patients more readily accepted why they might not be given an antibiotic right away for a respiratory tract infection/flu because of the consistent messaging and campaigns in this area, but these patients did not know that this message can also apply to UTI. One group was optimistic that the message about not receiving an antibiotic immediately was becoming more widely accepted. Various factors that would influence the clinician’s decision to provide an immediate antibiotic included renal function, co-morbidity, diabetes, immunosuppression, cost of treatment, pregnancy, severity of symptoms, age of patient, and duration of time before the weekend.

#### 3.4.4. Beliefs about Consequences/Memory, Attention and Decision Making/Knowledge

Most groups reported that they would discuss antibiotic resistance with some patients with a suspected UTI, especially if there were additional risk factors such as multiple courses of antibiotics. Some GP staff would discuss the need to complete the antibiotic course with patients, but they were less inclined to discuss antibiotic resistance in relationship to UTIs than other infections, especially if they were confident in their diagnosis. When indicating risk factors for antibiotic resistance, practices listed factors such as frequent/recent use of antibiotics, long duration or high doses of antibiotics, recurrent UTI, and positive laboratory results.

#### 3.4.5. Social Influence/Memory, Attention and Decision Making

Three focus groups reported challenges associated with only prescribing a 3 day course of antibiotics. Participants said these were often driven by patient expectations, since patients had received longer courses in the past and felt that the shorter dose would not be as effective, even though the cost is the same. Some prescribers also felt that the 3 day course was less effective than the longer one, with many patients returning with symptoms. This pressured them to lengthen the dose or use a different antibiotic.

#### 3.4.6. Knowledge

Some groups reported that they did not know about or have the option of using pivmecillinam or fosfomycin as second-line treatment for acute uncomplicated UTI and indicated this was because laboratories did not provide antibiotic susceptibility testing for these antibiotics.

#### 3.4.7. Behavioural Regulation

Some groups had audited their antibiotic prescribing, but none had audited antibiotic prescribing for UTIs specifically.

### 3.5. Self-Care and Prevention

#### 3.5.1. Skills/Knowledge

Most practices reported discussing self-management advice with their patients for mild symptoms (especially with a negative dipstick) or for preventing UTI recurrence. Key themes for prevention focused on hydration, cystitis sachets, cranberry products, lemon barley water, wiping front to back, and sodium bicarbonate (one group). There was confusion regarding the evidence for effectiveness of cystitis sachets and cranberry products for prevention or treatment of UTIs and whether these should be recommended. Hydration was discussed by most groups as a way to “flush out” the bladder or infection and was seen by some as the reason that cranberry juice is effective.

#### 3.5.2. Social Influence/Environmental Context and Resources

Some focus groups reported that patients might have received cystitis sachets from the community pharmacy and only came for a consultation if symptoms did not improve. Most groups indicated that more information for patients was needed, including a UTI leaflet with a public health campaign that covers the prevention and treatment of UTI, in line with national guidance.

## 4. Discussion

This study revealed practice and clinician variation in the use of urine dipsticks, empirical antibiotic choice and course length, exclusion of genitourinary causes and pyelonephritis, use of back-up/delayed antibiotics, and a lack of knowledge about the effectiveness of over-the-counter products for the prevention and treatment of UTI. There was also variation in the use of pivmecillinam and fosfomycin. Many UTIs were managed by nurse practitioners or by telephone consultation.

Our study found that clinicians reported using telephone consultations for the management of UTI. Small studies evaluating the use of protocol-led telephone consultations to manage UTIs show general success; however, a significant proportion of patients in these studies required physical examination, referral, or the need for urine culture because of the risk of complications [25,26]. A recent study in the United States of America (USA) assessed clinicians’ management of 20,600 telemedicine patients who were diagnosed with a UTI [27]. The study did not provide any guidance for clinicians regarding diagnosis and management of UTIs. Although the authors concluded that telemedical management was generally successful, there was an increase in the use of quinolones (against the USA’s national recommendations) in some groups and challenges with the management of patients with pyelonephritis, indicating that some clinicians would have benefited from simple decision-support tools. These studies and guidance highlight that face to face consultations are optimal, but if this is not possible there is a need to ensure that phone consultations follow an agreed diagnostic and management practice protocol that helps exclude other causes of urinary symptoms such as vaginal infection or atrophy, more severe infection like pyelonephritis, the initial signs of sepsis, and need for urine culture. Remote consultations in England have increased during the COVID 19 pandemic and digital-first primary care is part of the NHS Long-Term Plan [28,29]. Future diagnostic and management tools need to include guidance that can be applied to various types of consultations, including those done using phone or using other remote technology.

A recent UK study showed that overreliance on only urine dipsticks or clinical symptoms only may contribute to an under diagnosis of UTI, due to the limited ability of both to rule out infection [30]. Decision tools that use a combination of diagnostic symptoms and urine dipstick results might improve diagnostic accuracy and are currently the most reliable and cost-effective tools for the diagnosis of UTI [30,31,32,33,34]. However, clinicians need support to consider individual signs and symptom and recommend treatment options such as delayed antibiotics. Although many of the practice clinicians in our study followed national guidance to inform management, clinicians in our study still perceive pressure from patients to use urine dipsticks and prescribe based on their results. This patient expectation was perpetuated by ongoing use of urine dipsticks within the practice. Future UTI diagnostic guidance should include guidance on the role and effectiveness of urine dipsticks in detecting a UTI in all age groups, how dipstick results should be used alongside history and physical examination, and tools for communicating information on the appropriate use of dipsticks across the care pathway and with patients. UTI audits (that were not reported by any of our participants) with feedback and action planning may help embed good practice into daily practice care [35,36].

Non-steroidal anti-inflammatory drugs for UTI when used without back-up antibiotic prescribing for acute uncomplicated UTIs in women may lead to increased incidence of pyelonephritis [37]. However, NSAIDS used with the provision of a patient lead back-up antibiotic are acceptable to patients, and reduce antibiotic use without increasing complications [30,38,39]. Current national guidance includes recommendations for providing back-up antibiotics in women with uncomplicated acute UTI along with appropriate pain relief and advice about hydration [40]. Our findings indicated that GP staff felt that patients put pressure on clinicians to prescribe for a suspected UTI, and that this is often driven by the dipstick result and previous management experiences. This perception is similar to other qualitative research and could be related to lack of communication skills and knowledge about the consequences of infection, symptom resolution and risks and benefits of NSAIDS versus antibiotics [38,41]. Both patients and clinicians will need more information, evidence and discussion about consequences of overuse of antibiotics versus back-up antibiotics to motivate them to use back-up antibiotics more frequently for suspected UTI [38,41]. To improve public awareness, antibiotic campaigns and patient-facing leaflets could use language more inclusive of non-respiratory infections such as UTIs and highlight the benefits of back-up antibiotics. Continued effort should be put in to development of a more accurate point-of-care test that can more accurately guide the need for and choice of antibiotic [42].

Our findings, and others, show that at present, clinicians may not discuss non-pharmaceutical self-care and prevention of UTIs with women during a consultation and that some women do try over-the-counter products before seeking medical treatment [41]. National Institute for Health and Care Excellence (NICE) guidance now includes information as to the effectiveness of non-pharmaceutical products including cranberry, cystitis sachets and hydration for the prevention and management of UTI for pharmacy and GP surgery staff [40]. The NICE evidence review found little robust research to recommend cranberry products or other non-pharmaceutical means for the treatment of lower UTI, but did find that some non-liquid cranberry products could be helpful in the prevention of recurrent UTI for women under 65 [40,43]. A 2018 randomised control trial found that increasing water intake over a year could help reduce the number and frequency of cystitis episodes experienced by pre-menopausal women with recurrent UTI who previously drank less than 1.5 L of fluid a day [44]. However, there is little other robust evidence to demonstrate the effect of hydration on UTIs. Given the many non-pharmacological and non-antibiotic management options that are applied to all types of UTI, further research is needed to support recommendations and timing for non-pharmaceutical prevention of UTIs, hydration and “self-care” advice.

### 4.1. Other Studies

There are few qualitative studies assessing UTI management in primary care, with most focusing on treatment options, prescribing and antibiotic resistance rather than diagnosis of UTI. Qualitative data from an Irish study from 2015 highlighted the need for local antibiotic resistance surveillance data to inform antibiotic choice [45]. This Irish study and another in England found that patients want to discuss their illness and treatment options with their GP and are willing to accept a back-up/delayed prescription if time has been taken to explain the rationale for it [38,45]. A UK-based study, using qualitative data from GPs and female patients in 2011, showed that how a clinician manages a UTI depends on a complex mix of factors. Internal factors include personality, knowledge, teaching, and experience and external factors include perceived severity of diagnosis, guidance and protocols, training, cost of treatment, and perceived patient expectations/feelings [41]. A 2013 Swedish study indicated that GPs who most reliably follow treatment guidance for UTIs had a higher level of awareness and concern about increasing antibiotic resistance than less compliant GPs [46].

### 4.2. Limitations and Strengths

This study used stratification and random sampling techniques to select participants, which will have helped to limit bias. The sample size is also quite large when compared to other qualitative studies, allowing for a wider range of stakeholders with varying views. By using a known behavioural framework in the development of the focus group questions, we were able to ensure our interview schedule covered key discussion topics needed to understand behaviours around UTI management. This study was conducted alongside a programme of work to inform interventions to improve management of UTIs. This included a UTI flowchart and leaflet for older adults, which could have biased discussion to focus more on management in older adult patients, even when respondents were asked to provide feedback specific to younger age/all groups. To limit this, we asked for information about the general population or usual practice before prompting for information specific to older adults in the interview schedule. It is noted that more GP surgery staff from Gloucestershire participated in group discussions than from Nottingham (37 vs. 20, respectively). Gloucestershire CCG has almost twice the population of Nottingham City CCG and the average surgery list size is about 1300 people higher in Gloucestershire [47]. Because the strength of findings was based on consensus arising from group discussions (not comments from individuals), this should have limited any bias towards discussions with higher numbers of staff. Surgeries that were not able to participate indicated time constraints (often related to lack of staff/illness or changes within the surgery) and “no research” policies as a barrier to participation. This could have meant that our sample was biased towards practices that were more knowledgeable and interested in research and clinical issues like antimicrobial stewardship. However, topics related to antimicrobial stewardship were discussed at the end of the general discussion so should not have biased the initial conversations about UTI management in their practice. Further, there was a broad range of roles represented from across the practices, and we did reach data saturation.

## 5. Conclusions

This study highlighted variation in the care pathway for managing UTIs in adult women. Findings indicate this process could be improved by ensuring that guidance and protocols are appropriate for face to face consultations, as well as those by phone or using other remote technology, providing clear guidance on the effectiveness of urine dipsticks in detecting a UTI in all age groups. Practices should be encouraged to implement audits with feedback on UTI management. Tools to discuss management decisions and antibiotics with patients should be available across the care pathway; public campaigns around antimicrobial stewardship should use language more inclusive of non-respiratory infections such as UTIs. Further high-quality research is needed to provide additional guidance around non-pharmaceutical recommendations for “self-care” and prevention of UTIs.

## 6. Ethical Approval

As this study is part of service development, ethical approval was not required according to the Health Research Authority in England. However, this study was reviewed and approved by the Public Health England Research Ethics and Governance Group.

## Figures and Tables

**Figure 1 antibiotics-09-00581-f001:**
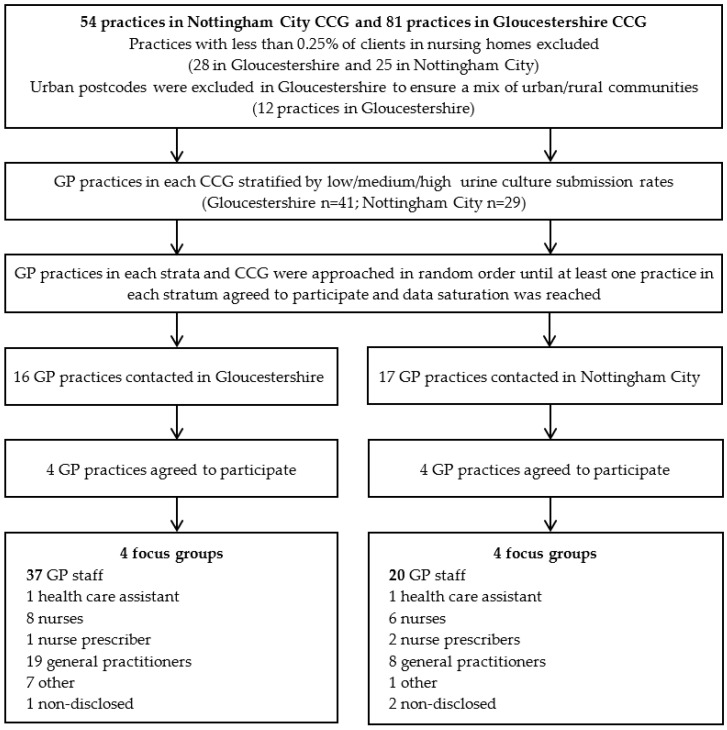
Sample selection.

**Table 1 antibiotics-09-00581-t001:** Demographic and health equality indicators for Nottingham City and Gloucestershire clinical commissioning groups (CCGs).

Indicators (2017/18)	Nottingham City CCG	Gloucestershire CCG	English Level Statistic
Total Population Estimates—mid 2018 [17]	331,069	633,558	55,977,178
Healthy life expectancy at birth for women in years	53.5 [18]	65.7 [19]	63.8 [18]
Life expectancy gap for women in years	8.6 [18]	5.4 [20]	7.5 [18]
% of all people with a limiting long-term illness	18.1 [18]	16.7 [19]	17.6 [18]
% of all people who are black and ethnic minority (2011 census) [16]	34.7	4.6	14.6
Deprivation score	36.9 [18]	15.0 [19]	21.8 [19]

**Table 2 antibiotics-09-00581-t002:** Interview discussion guide (summarized) and corresponding TDF domains.

Sections within the Interview Schedule	Main Domains Explored in the TDF
**Section One: Current UTI Management**	
1. Normal process for diagnosing a UTI in their surgery.	Knowledge/skills/social professional role and identity
2. Biggest challenges faced when diagnosing a UTI in their surgery?	Environmental context and resources/beliefs about capabilities
3. Differences in diagnostic process when assessing older and younger age groups.	Knowledge/skills/beliefs about consequences
4. Determining the need for antibiotics for a suspected UTI and how easy it is to work out.	Memory, attention and decision making/beliefs about consequences/social influence/beliefs about capabilities
5. Determining if urine is collected or sent for culture.	Memory, attention and decision making/social influence
6. How instructions are given on how and when staff or patients should collect a urine sample.	Behavioural regulation/social professional role and identity
7. How a practice monitors and regulates the diagnosis and management of UTIs (including antibiotic use).	Environmental context and resources/reinforcement
8. Barriers to diagnosing suspected UTI in different groups including the influence of family and caregivers.	Environmental context and resources/social influences/emotions
9. Discussing benefits vs risks of antibiotics with patients.	Skill/intentions/goals
10. Views/experiences of antibiotic resistance in relation to UTIs.	Memory, attention and decision making/beliefs about consequences/behavioural regulation
11. Discussing antibiotic resistance with patients/caregivers.	Knowledge/skills/beliefs about consequences/intentions
12. Ways to reduce the number patients who are prescribed antibiotics (role specific).	Optimism/social professional role and identity/motivation/goals
**Section Two: Ways we could reduce UTIs or improve UTI management**	
1. Diagnostic or treatment guidance currently used to manage patients with a suspected UTI.	Environmental context and resources/behavioural regulation/reinforcement
2. Support/resources that would assist in identification and diagnosing UTI.	Skills/environmental context and resources
**Section Three: How a UTI resource might be used to improve diagnosis and treatment** (corresponds with a step by step review of the previous or updated UTI diagnostic resources)	
1. Information in a reference tool that would help with UTI prevention, diagnosis and management.	Knowledge/memory, attention and decision making/skills
2. Discuss the usefulness of this type of resource.	Environmental context and resources/memory, attention and decision making/beliefs about capabilities
3. Discuss the layout and wording used in the diagnostic tool.	Knowledge/memory, attention and decision making/skills
4. Discuss what would make a quick reference tool like this more attractive or usable for practice staff.	Environmental context and resources
5. Discuss challenges in using a resource like this diagnostic tool.	Beliefs about capabilities/reinforcement

**Table 3 antibiotics-09-00581-t003:** Key findings and how they influenced the development of the UTI diagnostic tool for women under 65 years. (Results (Applied to the TDF Domains)).

**Domain and Findings ***	**Quote** *(F Stands for Female, M for Male, Int Stands for Interviewer and GP Stands for General Practice)*	**How the Findings Influenced the UTI Diagnostic Tool**
**Knowledge**		
Some general practice staff reported that they did not know about pivmecillinam or fosfomycin as second-line treatment for acute uncomplicated UTI.	*“F1: I’ve never even heard of it before (pivmecillinam). What’s it called…? F3: Not seen it at all. M1: I’ve not seen that at all.” (GP 8)*	The updated diagnostic tool: Links to latest national guidance on antimicrobial prescribing for UTIs; Lists the main at-risk groups and other risk factors to consider for antibiotic resistance, and Links to a UTI leaflet for women under 65 years that explains evidenced-based prevention and self-care recommendations.
GP practices discussed frequent/recent use of antibiotics, long duration/high doses of antibiotics, recurrent UTI, and positive lab results as risk factors for antibiotic resistance.	*“Certainly, the people coming back, recurrent UTIs… you sometimes think are we actually missing something here are they having proper UTI, the dipstick might show it but actually having an MSU with a proven UTI can help in the long term, so that’s one group.” (GP 5)*	
Practice suggestions for prevention of UTI focused on hydration, sachets/cranberry products, lemon barley water, and wiping front to back.	*“M1: Occasionally Cystopurin. F1: …the sachets. Especially if the dipstick’s negative. M1: I’d even recommended sodium bicarbonate… F2: It used to be barley water in my day…. F3: Basically just get as much fluid down you as you can and flush it out.” (GP 6)*	
Some GP staff discussed conflicting information regarding the use of cystitis sachets and cranberry products for prevention or treatment of UTIs.	*“Cranberry juice seems to have fallen out of fashion. The wording… in the guidance now… is that cranberry juice is effective for some women, but there is very limited research to back it up. the impression that we get is that the reason why cranberry juice is working is because they’re drinking a lot of fluid.” (GP 1)*	
**Environmental context and resources**		
Some GP staff reported that phone consultations (sometimes with dipstick results) are being used to assess and treat patients at risk for a UTI.	*“…we ask them relevant questions, how long they’ve had the symptoms for, have they tried any over the counter, what are their symptoms, whether it’s dysuria, frequency, back pain, temperature, nausea, vomiting, that sort of thing, and it depends on the answers… particularly at our surgery, if they’re under 65, we just, they give them a prescription over the phone.” (GP 1)*	The updated diagnostic tool: Clearly outlines how to clinically assess someone with suspected UTI; Includes prompts/considerations around differential diagnosis, pyelonephritis, and sepsis; Clearly outlines the steps for clinical assessment of someone with suspected UTI and when a urine dipstick test or culture is needed; Provides information on the sensitivity and specificity when using urine dipsticks to diagnose a UTI for women under 65 years; Has been developed as an update to previous guidance; Links to latest national guidance on antimicrobial prescribing for UTI management (developed since this study was conducted); Links to UTI leaflets and resources for women under 65 years that explains evidenced-based prevention and self-care recommendations.
Some groups reported that they would prescribe antibiotics over the phone for a patient under the age of 65 with 2–3 urinary symptoms.		
At the telephone consultation, some practice staff asked for a urine sample to be brought in but others considered this as unnecessary.	*“Int: how would you diagnose a suspected UTI over the phone versus in face to face consultation? F1: Well just do the same but not examine them or dip it… we tend to not ask for a sample unless there was something complicated about them.” (GP 3)*	
Some practice staff reported that patients ‘drop and run’, leaving urine samples at reception, often because they think they might have a UTI. Clinicians then ring the patient for a clinical history.	*“M1: …we do end up with patients just turning up and leaving urine samples at the desk. Int: So what determines whether they get sent for culture? M1: Well, well they get dipped but then we want to know a bit more of the clinical information so… the nurses comes and says, I’ve checked this urine. We need a bit more than that… I think commonly we’d just end up phoning the patient and saying look… why have you dropped it off?” (GP 2)*	
Some GP staff feel that easy availability of urine samples influences their decision to use urine dipsticks as part of the diagnostic process.	*“…think for the vast majority we would test… even in females under 65 who… you certainly don’t need to send a culture for but if a urine is easily obtainable we’d normally get one.” (GP 2)*	
Most GP staff followed 2017 PHE guidance for UTI diagnosis and which focused on the key symptoms of dysuria, urgency and frequency as symptoms and would not include findings from the most recent literature.	*“…and I use the current guidance to decide whether to treat without dipping the urine or to just go ahead and treat. And I think it’s if they have three symptoms, that you don’t have to dip the urine and you just treat. If they’re under 65 and they’re a woman then I would tend to use that….” (GP 2)*	
Most GP staff agreed that the exclusion of pyelonephritis and sepsis needed to be included in tools that are developed to diagnose UTI.	*“I think sepsis has to be first, doesn’t it? It’s the trump card, is it?” (GP 4)* *“…and obviously, you’ve got to think about sepsis now.” (GP 6)*	
Some GP staff expressed the need for more national guidance specific to the management of UTIs.	*“…something on recurrent UTI would be really helpful because… some guidance… just sort of says consider this, consider that… there’s a lot more interpretation with recurrent UTI.” (GP 2)*	
Some GP staff reported that lab specimen service pickups were not frequent enough to allow culture results to inform treatment, which leads to overcautious prescribing.	*“…I’m going to treat because I won’t get the result of this MSU, this is a Thursday I’m not going to get it till Tuesday so we tend to treat.” (GP 7)*	
Some GP staff reported that they did not have the option of using fosfomycin or pivmecillinam as second-line treatment.	*“F1: But they didn’t actually have fosfomycin or pivmecillinam on the (sensitivity reporting) list. M1: No they don’t have fosfomycin….” (GP 2)*	
Most GP staff indicated that more information or education for patients on UTI was needed, including a UTI leaflet and public health information/campaign that covers the prevention and treatment of UTI in line with national guidance.	*“…like with coughs and colds everyone knows now that they’re not going to get an antibiotic necessarily, they’ve braced themselves for the bad news, haven’t they? But with UTI they wee once and it stings and… they separate it and somehow it doesn’t go under that hurt. It’s literally they’re right in, aren’t they, really quickly.” (GP 8)*	
**Role and professional identity**		
Some GP staff reported that receptionists in some surgeries ask patients who ring up to bring urine samples into the surgery and provide guidance on sample collection.	*“The receptionists actually, if anybody rings up and says, I think I’ve got a UTI they, they say can you, next time you go can you make sure you keep a clean catch? So that we’ve already started a process of looking at having a… dip if needs be.”* (GP 2)	The updated diagnostic tool: Can be displayed in a surgery so that staff members can access it; Uses clear language that can be understood by different types of health professionals; Outlines the steps for clinical assessment and when a urine dipstick test/culture is needed; Provides information on the sensitivity and specificity of urine dipsticks to diagnose a UTI for women under 65 years; Includes prompts/considerations around other diagnosis, pyelonephritis, and sepsis; Links to national sepsis guidance;Includes information on when to refer patients who have recurrent UTI or other significant symptoms.
Some GP staff reported that nurses may be responsible for testing urine samples left at reception and referring to clinicians if they are concerned the patient has a UTI.	*“…often, they’ve already had a urine dipped by our nurses by the time that they get to us…. If the other nurses see someone and they’re suspecting a UTI they’ll… ask us to prescribe, but they’ll tell us what they’ve found, examination wise and history. So run it by us.”* (GP 2)	
Some GP staff reported that nurse prescribers frequently see uncomplicated cases of UTI independently.	*“…I (nurse prescriber) deal predominantly in the triage and assessment of acute illnesses and UTIs being probably quite an awful lot of it and it’s part of a team where we prescribe quite regularly for it following national guidelines.”* (GP 8)	
Some GP staff reported that general practitioners will deal with UTIs but often focus on complicated cases or treatment failures.	“*I tend to be seeing people who either they’ve got complicated features or they’ve had treatment already, or there’s something else…. I do quite a bit of telephone work, advice work.”* (GP 1)	
**Skills**		
Some GP staff asked about vaginal discharge during a consultation, and one group reported this was done after using a urine dipstick to rule out a UTI.	*“…you ask for certain symptoms… if it sounds like a UTI and it seems like a UTI on a dipstick you treat as such. It would only be if you weren’t sure, if you needed to think about other symptoms then you would say have you got a vaginal discharge.” (GP 7)*	The updated diagnostic tool: Flags the need to assess for vaginal discharge early in the diagnostic process; Suggests when to use urine dipsticks; States criteria for sending urine for culture; Links to a UTI leaflet for women under 65 years that provides evidenced-based recommendations for self-care, prevention, and safety netting.
Most groups said they would send a culture following treatment failure or for a recurrent UTI but only some groups reported sending a urine for culture if the dipstick was negative but the patient symptomatic, if the patient was quite unwell, for male patients, pregnant women and for children.	*“…I don’t use an age cut off. It’s more around their past history, for instance if they’re pregnant or something. Or if their symptoms aren’t very clear cut, or if they’re unwell or they’ve had recurrent then I will send off…. If it’s a treatment failure or they come back and their symptoms aren’t fully cleared up, second time round, then certainly we’d send it off at, on the second appointment if, if needed, if they come back...” (GP 3)*	
Some groups indicated that they would not send a urine culture if the dipstick was negative or if the symptoms clearly indicated an uncomplicated UTI.	*“With different things depending on the patient. So a pregnant or recurrent or, we’re more likely to send a urine off. We don’t send them off if it’s an uncomplicated UTI with typical symptoms.” (GP 3)*	
Most GP staff reported discussing non-medical advice with their patients. This was in the context of self-care for mild symptoms (especially with a negative dipstick), or preventing UTI recurrence.	*“Yeah, but if they’ve got relatively mild symptoms or few symptoms you can let them carry on flushing it out…” (GP 2)*	
Some GP staff reported differences of opinion on interpretation/use of urine dipstick; especially when symptoms do not match urine dipstick results which challenge UTI diagnosis.	*“…people with recurrent symptoms… doesn’t always match up the symptoms that they have with the findings in their urine and possibly their expectation for treatment.” (GP 2)*	
**Social Influence**		
Some GP staff feel that patient expectation influences their decision to use urine dipsticks as part of the diagnostic process.	*“…if they’re young, fit, well, uncomplicated and got typical symptoms, technically you don’t need to do a dipstick. But you often still do it because the patients have that expectation because they want to be told this is definitely a urine infection. If you just say, you’ve got typical symptoms here are antibiotics, they still feel as though they want to be told this is an infection, so they will ask for it to be dipped or expect it to be dipped.” (GP 3)*	The updated diagnostic tool: Links to a UTI leaflet for women under 65 years that provides support to a clinician when discussing with a patient about: o When a dipstick result should be used to diagnose a UTI; o What a delayed prescription is and when to use it; o Safety netting advice; o Evidenced-based self-care and prevention advice for UTIs, including over-the-counter products.
Some practices generally felt that patients did not understand that a UTI can clear on its own safely and expected treatment and believe that patients react differently to a no antibiotic decision with respiratory infections due to more consistent messaging/campaigns in this area.	*“Almost never, usually, if it’s for something like a sore throat or what have you, then… they’re getting more receptive, because it’s been a consistent message that they’re not going to get antibiotics unless X, Y, Z, but in urinary tract infections almost never, they just want them.” (GP 4)*	
Some practices talked about how patients might obtain cystitis sachets from the community pharmacy and will only come for a consultation if symptoms don’t improve.	*“…because we do get patients who have been to the pharmacist and I don’t see the ones who don’t come back, but I do see the ones who do, and… after week later they’re now going their urine is painful and they need antibiotics….” (GP 5)*	
**Beliefs about consequences**		
Some GP practices indicated that concern about issues like sepsis causes them to be overly cautious when diagnosing and treating UTIs.	*“…it’s confidence around knowing… what happens if we don’t treat it, there is that risk or pyelonephritis and becoming septic… because of this lack of confidence we’ve always treated UTIs so I think it’s a number of different factors that apply.”* (GP 7)	The updated diagnostic tool: Clearly outlines the steps for clinical assessment of someone with suspected UTI; Includes prompts/considerations around differential diagnosis, pyelonephritis, and sepsis; References national sepsis guidance; References national guidelines on delayed and back-up prescribing for UTIs; Clarifies criteria and risk factors for antibiotic resistance; References recent evidence; around hydration as a mechanism for preventing recurrent UTI.
Most GP staff would not usually use a delayed/back-up prescription for UTIs.	*“Usually not so much UTIs because by then they’ve already had the symptoms they’re coming in and they’re symptomatic, whereas a lot of people with sore throats come in and say it started yesterday. So, you can give them delayed, but I’m sure the percentage that cash in a delayed prescription is a lot higher” (GP 7)*	
Most groups reported that they would discuss antibiotic resistance with some patients with an expected UTI, especially if there were additional risk factors.	*“Occasionally, gradually, particularly if they’ve got a recurrent UTI or if they’ve had, say, side effects following antibiotics previously, like thrush. Then you might have a conversation about the pros and cons.” (GP 3)*	
Hydration was discussed by most groups as a way to “flush out” the bladder/infection and was seen by some as the reason that cranberry juice is effective.	*“Basically, just get as much fluid down you as you can and flush it out. Because if the urine analysis is negative each time, …that’s all you can advise them.” (GP 6)*	
**Beliefs about Capabilities**		
Though most practice staff used national guidance to manage UTIs, some would not refer to it every time.	*“I intermittently look at the guideline and make sure what I think and I’m doing is correct, and then invariably I look and realise what I think I’m supposed to be doing is slightly changed over the couple of months since I last read it.” (GP 2)*	The updated diagnostic tool: Is developed to be integrated into GP systems so that it can be updated and referred to easily.
**Memory, Attention, and Decision Making**		
Some GP staff believe that 3 day courses of antibiotics were not sufficient. This was driven by patient expectations and perceived efficacy of the dose and pressured them to lengthen the dose or use a different antibiotic.	*“M1: …I still find it quite hard to prescribe in three days, (some agreement) because patients don’t like three days. M2: They don’t do they, they don’t think it’s worth it… M3: They do not think that you’re giving them enough and that’s quite tricky I think, I still find that, trying to convince them that three days will be enough.” (GP 7)* *“I never find that three days of antibiotic works. I get so many people coming back (with UTI symptoms) that I don’t prescribe three days any more actually.” (GP 3)*	The updated diagnostic tool: References national prescribing guidelines for UTIs which clearly states risk factors and co-morbidities that need to be considered when treating a UTI; References national guidelines on delayed and back-up prescribing for UTIs; Clarifies criteria and risk factors for antibiotic resistance; Links to a UTI leaflet for women under 65 years that provides support to a clinician when discussing with a patient: o What a delayed prescription is and when to use it o Risk related to antibiotic use and resistance.
Some GP staff would delay antibiotics when waiting for results from urine culture if symptoms were mild.	*“It depends on the day of the week, so if it’s a Thursday and the results aren’t going to come back, I might give them a prescription and say let me send it, if you get worse whilst you’re waiting for the results, that’s the context that I would do that.” (GP 3)*	
Some GP staff would discuss completing the antibiotic course but were less inclined to discuss antibiotic resistance in relationship to UTIs than other infections, especially if they were confident in their diagnosis.	*“I think it also depends on how convinced I am they’ve got a wee infection. So, if I’m convinced they’ve got a wee infection, and they’re quite systemically unwell with it, then I may not fight trying to give them the antibiotics as much as I might do if they’re going: I’ve had a bit of dysuria and I don’t know quite what’s going on.” (GP 4)*	
Various factors that would influence the clinician’s decision to provide an immediate antibiotic include renal function; co-morbidity; diabetes; immunosuppression; cost of treatment; pregnancy; severity of symptoms; age of patient; and time before the weekend.	*“EGFR, their diabetes status, the recurrence of the symptoms, whether they’re pregnant or not, that sort of thing, so there’s quite a few other issues that I need to then take on board despite the guidance that’s being given out.” (GP 8)*	
**Optimism**		
One GP group was optimistic that the message about not receiving an antibiotic immediately was becoming widely more accepted.	*“People are starting to get that message now, with what’s been in the press, …it’s refreshing that you now come across more patients who will say… I don’t want one if I don’t have to have one. And if you say, let’s give this a few days or let’s wait for the culture to come back or whatever and see, they’re OK with that.” (GP 2)*	The updated diagnostic tool: Links to a UTI leaflet for women under 65 years can be used by a clinician when discussing with a patient about the need for antibiotics and alternative options for UTI management.
**Behavioural Regulation**		
The usefulness of a having an accurate point-of-care test was mentioned by some groups, but the perceived benefit was often to provide reassurance to the patient.	*“…oximetry, one thing I do with these people the ones who come with cold and cough, I put them on the machine, I say look, oxygen, pulse is level, temperature normal, you haven’t got an infection and they love it, so that reassures them. If there was a something which we got, where we put the urine and say look, it says there’s no infection. They will be happy.” (GP 6)*	The updated diagnostic tool: Links to the TARGET Antibiotics Toolkit which provides tools to audit UTI prescribing.
Some groups had audited their antibiotic prescribing recently and none had audited antibiotic prescribing for UTIs specifically.	*“Different members of staff have audited different topics, haven’t you? We’re doing one at the moment… on COPD antibiotics. We have our own internal audits and most of you do an audit for your appraisals and things as well don’t you?” (GP 7)*	
**Reinforcement**		
Most GP staff believed that patients expect to be prescribed an antibiotic if they have urinary symptoms	*“M1: I think, as *** has said, it’s patient expectations. F1: So they have these symptoms and they’ve always had antibiotics. M1: They’ve had the symptoms. F1: So they need antibiotics again, or they’ve got the same, the same symptoms and don’t often consider that there might be another, another cause.” (GP 4)*	The updated diagnostic tool: Links to a UTI leaflet for women under 65 years that provides support to a clinician when discussing with a patient about: o What a delayed prescription is and when to use it o Risk related to antibiotic use.

There were no significant findings that were more related the domains of goals, intentions, and emotions than others covered above. * The terms used to attribute group consensus in the results include “most”—4 or more focus groups agreed/discussed; or “some”—1 to 3 groups agreed/discussed.

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
