# Peer review of "Diagnosis and Management of UTI in Primary Care Settings—A Qualitative Study to Inform a Diagnostic Quick Reference Tool for Women Under 65 Years"

_antibiotics, 2020, doi:10.3390/antibiotics9090581_

Round 1

Reviewer 1 Report

Overall the manuscript discusses an important issue, which is variation in prescribing practice of antibiotics for UTI. Main comments:

1) The specific gap in the literature needs to be more clearly highlighted in the introduction, and the manuscript is rather long and due to its qualitative approach. The key findings were only much more clear in the discussion.

2) The low response rate is a concern, but this was addressed as a limitation. 

3) Some parts such as Table 3 can be made much more clear and concise.

4) Was there an ethics statement included in the main text body? If so, it was missed.

Specific comments are noted further on the manuscript pdf. 

Author Response

Overall the manuscript discusses an important issue, which is variation in Thank you prescribing practice of antibiotics for UTI. Main comments:

Thank you for your excellent feedback.

1) The specific gap in the literature needs to be more clearly highlighted in the introduction, and the manuscript is rather long and due to its qualitative approach. The key findings were only much more clear in the discussion.

I included some points in the background specific to what you raised in the PDF. Specifically why we have focused on women under 65 when this was not stated in our aims and why England still needs to address UTI management as a means to improve prescribing

2) The low response rate is a concern, but this was addressed as a limitation.

Yes - I have done nothing more to note this.

3) Some parts such as Table 3 can be made much more clear and concise.

I have reduced the length of table 3 by editing the text of the comments and re-organising the tables. It has reduced from 8 to 6 pages. I hope the new lay out makes it clearer that we are trying to evidence how the findings influenced the flowchart development.

4) Was there an ethics statement included in the main text body? If so, it was missed.

I included a statement specific to ethics at the end of the document.

Specific comments are noted further on the manuscript pdf.

Thank you, I have included responses to each of your comments within the PDF, but have only made changes in the updated word document.  Some comments were blank and I didn't respond to these.

Reviewer 2 Report

The manuscript is interesting and sounding, reporting important informations about UTIs management in specific settings.

Please revise the reference numbering from ref. #2

Author Response

Thank you for your feedback.

I have edited the references.

Reviewer 3 Report

I read this paper with a really great interests. World Health Organization (WHO) recognizes currently antibiotic resistance as one of the biggest threats to global health. According to the data published by Centers for Disease Control and Prevention each year in the U.S. more than 35 000 people die due to antibiotic resistance infections.WHO recommends to the professionals to “only prescribe and dispense antibiotics when they are needed”. The authors of this paper reported that Escherichia coli is the main cause of bloodstream infection in the UK and is responsible for more  than one third of blood stream infection cases in England each year. As a gynaecologist I see in a daily practice a numerous number of women suffering from UTI. At least 50% of women will develop one UTI episode during life. Recurrent urinary tract infection is defined as ≥2 infections in six months or ≥3 infections in one year. It has been observed that 27 percent of women will develop at least one culture-confirmed recurrence within the 6 months following the initial infection and therapy. Therefore, effective non-antibiotic approach in UTI treatment or prophylaxis remains very attractive for both as physicians and patients.

The authors concluded "Public antibiotic campaigns and patient facing information should cover UTIs, non-pharmaceutical recommendations for “self-care”, prevention and rationale for 3-day antibiotic courses and I absolutely agree with this statement. Therefore I strongly recommend this paper for publication.

Author Response

Thank you for the information below. We are glad to contribute to knowledge in this area.

Reviewer 4 Report

This is a very nice study regarding the approach of GP practices in the UK to urianry tract infections in females under 65 years of age. It highlights the problems that UTIs or symptoms that could be associated with a UTI pose for clinical management. In their comprehensive survey of GP practices they determined experience with this entity, diagnostic test use (and their challenges) and antibiotic use and were able to develop recommendations especially in regards to antibiotic stewardship.

Author Response

Thank you for your comments.